# The role of 5-HTTLPR in autism spectrum disorder: New evidence and a meta-analysis of this polymorphism in Latin American population with psychiatric disorders

D. L. Nuñez-Rios[1], R. Chaskel[2,3], A. Lopez[3,4], L. Galeano[1], M. C. Lattig [1,4]*

1 Departamento de Ciencias Biológicas, Universidad de los Andes, Bogotá, Colombia, 2 Instituto Colombiano del Sistema Nervioso Clínica Monserrat, Bogotá, Colombia, 3 Fundación Santa Fe de Bogotá, Bogotá, Colombia, 4 Servicios Integrales en Genética (SIGEN) alianza Fundación Santa Fe de Bogotá – Universidad de los Andes, Bogotá, Colombia

* mlattig@uniandes.edu.co

## Abstract

The autism spectrum disorder (ASD) is a complex disorder encompassing a broad phenotypic and genotypic variability. The short (S)/long (L) 5-HTTLPR polymorphism has a functional role in the regulation of extracellular serotonin levels and both alleles have been associated to ASD. Most studies including European, American, and Asian populations have suggested an ethnical heterogeneity of this polymorphism; however, the short/long frequencies from Latin American population have been under-studied in recent meta-analysis. Here, we evaluated the 5-HTTLPR polymorphism in Colombian individuals with idiopathic ASD and reported a non-preferential S or L transmission and a non-association with ASD risk or symptom severity. Moreover, to recognize the allelic frequencies of an under-represented population we also recovered genetic studies from Latin American individuals and compared these frequencies with frequencies from other ethnicities. Results from meta-analysis suggest that short/long frequencies in Latin American are similar to those reported in Caucasian population but different to African and Asian regions.

**Data Availability Statement:** All relevant data are within the paper and its Supporting Information files.

## Introduction

Autism Spectrum Disorder (ASD) is a neurodevelopmental condition characterized by three core symptoms: repetitive/restricted behaviors, impairment in social interaction and variable communication skills [1]. The twin concordance rate and a milder autism phenotype in relatives reflects a strong genetic component in the pathophysiology of ASD; nevertheless, the broad phenotypic variability of this disorder and the 60% of individuals who remain with unknown etiology suggest an interplay of several genetic factors [2–4]

Most of the genetic factors involved in ASD have a role in brain development and the excitatory/inhibitory synaptic balance [5–8]. The serotoninergic system participates in neurogenesis, axon guidance and cell migration, and also modulates GABA and glutamate

**Funding:** MCL: Grant # 120474455837 (744-2016) from Minciencias, previously known as Colciencias MCL: Grant Heterogeneidad Genetica del Autismo. Vicerrectoria de investigacones -Universidad de Los Andes DLN: scholarship from Ceiba.

**Competing interests:** The authors have declared that no competing interests exist.

neurotransmitter release in presynaptic terminals. Within the serotoninergic system, the serotonin re-uptake transporter (SERT) located in presynaptic terminals and encoded by the *SLC6A4* gene (Solute carrier family 6 member 4, Gene ID: 6532) has been widely studied. The transcriptional efficiency of *SLC6A4* is regulated by the well-known short (S)/long (L) 5HTTLPR (5-HTT gene-linked polymorphic region) polymorphism, a repetitive sequence present in the upstream regulatory region of this gene [9–14]. The short allele which reduces the transcriptional activity has been reported with greater frequency in African and Egyptian individuals with ASD [15,16], and has been also associated with other psychiatric disorders such as major depressive disorder [17–20], bipolar disorder [21] and depression risk in Parkinson disease [22] among others; however, conflicting results have been also reported [23–25].

Despite three meta-analysis suggesting that the short/long alleles are not risk factors for ASD [26–28], the ethnical heterogeneity among included studies has been proposed as a factor that may affect the overall result. A comprehensive transmission disequilibrium test (TDT) meta-analysis including eighteen studies from diverse populations, demonstrated a non-preferential S/L transmission at the global scale, but preferential transmission of the S and L alleles in the American and Asian populations, respectively [29]. Thus, based on the frequency differences of 5-HTTLPR polymorphism in the worl-wide population, here we present an attempt to understand the role of short/long allele in Colombian population with idiopathic ASD in terms of risk, preferential transmission and symptoms severity. Additionally, we performed a meta-analysis aiming to evaluate the heterogeneity of the S and L alleles in the under-represented and highly admixed Latin America population.

## Materials and methods

The research protocol was approved by ethics committee of participating institutions (Universidad de Los Andes and Instituto Colombiano del Sistema Nervioso—Clínica Montserrat). Written informed consent was obtained by all participants under 16. Parent or legal guardian written consent was obtained for all participants 15 and under or unable to consent.

### Subjects

Idiopathic ASD diagnosis confirmed in 105 individuals was performed using the Diagnostic and Statistical Manual of Mental Disorders V (DSM-V) criteria [1]. Clinical evaluation also included the completion of the ADOS [30,31] and ADI-R [32] instruments. For our case-control analysis we used genotypic information of 171 unrelated and unaffected Colombian individuals belonging to the same region as the ASD trios. Peripheral blood was obtained from all participants and parents. DNA was extracted using Flexigene® DNA kit (Quiagen, Inc).

### Genotyping

5-HTTLPR polymorphism was screened through PCR and gel electrophoresis according to Petri S et al., 1996 protocol [33]. PCR product was visualized with 3% MetaPhor agarose gel electrophoresis [Lonza Group Ltd., Basel, Switzerland].

### Literature search

The Meta-analysis followed the Preferred Reporting Items for Systematic Reviews and Meta-analysis (PRISMA) criteria [34]. We screened without date restriction reports that evaluate the association of 5-HTTLPR and psychiatric disorders in PubMed, ScienceDirect and Scielo databases up to April 2020. To minimize the chance of missing relevant studies the search terms in PubMed and ScienceDirect databases were "HTTLPR" "SLC6A4" AND the name of each

country encompassed in South and Central America (each separated by the Boolean operator OR). For the Scielo database, we searched published literature in the indexed journals from Latin American countries (S1 Fig). The inclusion criteria were (1) full text article in English or Spanish languages (2) case-control studies evaluating the polymorphism in psychiatric disorders (3) genotypic or allelic frequency provided and (4) control individuals not deviated from Hardy-Weinberg Equilibrium (HWE). We excluded reviews, studies evaluating trios, cases and no controls, population without a clinical diagnosis and studies that did not evaluate the Latin American population or the association of 5-HTTLPR with psychiatric disorders. For articles that did not have all the information available the authors were contacted. The data extraction was carried out by two investigators independently. The collected data from each study was categorized as reference, country of origin, evaluated trait, number of individuals and allelic/genotypic frequencies.

## Statistical analysis

The test of association between ASD and 5-HTTLPR polymorphism was evaluated under the allelic (L vs S), genotypic (SS vs SL vs LL), dominant (LL/LS vs SS) and recessive (SS/SL vs LL) models using the genotypic information of 105 individuals with ASD and 171 unaffected/unrelated controls; corrections were conducted by the Bonferroni method. Linear regression and quantitative trait association modeled the severity of common ASD traits as a function of the 5-HTTLPR genotype. The family-based association for the 5-HTTLPR genotype and ASD was evaluated with the transmission disequilibrium test (TDT) using genotypic information of 105 trios. For the meta-analysis, the allelic and genotypic frequencies were included as a common measure to evaluate the association between 5-HTTLPR polymorphism and psychiatric disorders in Latin American population. The heterogeneity across studies was estimated by the Cochran's Q test and the pooled Odds Ratio (OR) was evaluated under allelic (L vs S) and genotypic (SS vs LL+LS)—(LL vs SS+SL) models with fixed-effect or random-effect models according to $I^2$ value (Fixed-effect for $I^2 <50\%$ and Random-effect for $I^2 > 50\%$). The publication bias was assessed with funnel plots and quantitatively evaluated with Egger's regression and Begg's rank correlation. The trim and fill method was used to estimate potential missing studies and the sensitivity analysis removing each study for every meta-analysis was conducted to evaluate the stability of the results. All statistical analysis were performed with PLINK [35] and R studio program [36].

## Results

The genotyping results of 105 trios and 171 unrelated controls are described in S1 and S2 Tables. The genotypic and allelic frequencies are summarized in Table 1 and full sample

**Table 1. 5-HTTLPR polymorphism in Colombian individuals.**

| Group | Genotype frequency | | | | Allele frequency | |
|---|---|---|---|---|---|---|
| | LL | SL | SS | Total | L | S |
| Father | 19 [18%] | 49 [47%] | 37 [35%] | 105 | 0.41 | 0.59 |
| Mother | 23 [22%] | 42 [40%] | 40 [38%] | 105 | 0.42 | 0.58 |
| Child with ASD | 22 [21%] | 42 [40%] | 41 [39%] | 105 | 0.41 | 0.59 |
| Unrelated controls | 38 [22%] | 89 [52%] | 44 [26%] | 171 | 0.48 | 0.52 |

Genotypic and allelic frequencies in ASD trios and unaffected individuals

**Table 2. Phenotypic traits.**

| ADOS score in investigated dimension [Mean ± SD] | SS | SL | LL | *p*-value |
|---|---|---|---|---|
| ADOS Social communication | 5,6 ± 1,9 | 5,9 ± 1,8 | 5 ± 1,8 | 0.4936 |
| ADOS Restricted/Repetitive behaviors | 2,9 ± 1,6 | 2,8 ± 1,3 | 2,9 ± 1,6 | 0.964 |
| ADOS Social interaction | 8,4 ± 3 | 9 ± 2,7 | 8,2 ± 2,4 | 0.8907 |

Mean score and standard deviation to each trait is presented according to genotype.

analysis including families and unrelated controls were consistent with frequencies predicted by Hardy-Weinberg equilibrium ($p = 0.12$).

Although a significant association between 5-HTTLPR and ASD was observed only under dominant model (SS vs SL+LL, ($p = 0.022$)) (S3 Table), after Bonferroni's correction non-significant result was observed. Additionally, the transmission disequilibrium test (TDT) indicates a non-preferential transmission of either the short or long allele ($\chi^2 = 0.0989$, $p = 0.75$) (S4 Table).

Phenotypic features and comorbidities in individuals with ASD were analyzed according to genotype (S5 Table). Aggressive behaviors were the most common comorbidity reported followed by intellectual disability and epilepsy. The sex ratio was 6:1 (male-to-female) with mean age of 10,8 ± 7,8 years old. For reasons of statistical power, the analysis between 5-HTTLPR and autism severity was restricted to the three core ASD symptoms (S1 and S2 Tables). The mean score for each trait was similar among the three genotypes and statistical test confirmed a non-association (Table 2).

A total of 52 publications with the keywords provided in PubMed database were initially filtered, but only 13 case-control studies met the inclusion criteria: two from Colombia, three from Mexico, one from Argentina and seven from Brazil. In ScienceDirect database we found an additional study from Brazil and in Scielo database we found four additional studies: two from Colombia, one from Mexico and one from Brazil. No response was obtained from the authors that were contacted. Thus, from a total of 112 filtered reports, only 18 fulfilled the inclusion criteria (S1 Fig), and the allelic frequencies of each study were separately registered for cases and controls (Table 3).

The Latin American meta-analysis performed under three models (S vs L), (SS vs SL+LL) and (LL vs SL+SS) reflected a heterogeneity of 33.1%, 19.5% and 13.7%, respectively (Table 4, Fig 1 and S2 Fig). Fixed-effect model was selected to estimate the pooled OR based on heterogeneity results. The overall OR in either of the three models failed to find significant association (Table 4) suggesting that the 5-HTTLPR polymorphism does not increase the risk for psychiatric disorders in Latin American population (Fig 1 and S2 Fig). For publication bias, the funnel plots suggested absence of a bias (S3 Fig) and both Egger´s and Begg's test confirmed non significance (Table 4). The trim and fill method did not identified missing studies for each model (S4 Fig) and the recalculated OR did not yield different conclusions (Table 4) just like the sensitivity analysis (S6 Table).

## Discussion

The serotonin re-uptake transporter (SERT) located in presynaptic neuron removes this neurotransmitter from the synaptic cleft, regulates the serotonin concentration in the synapse and controls the magnitude or duration of post-synaptic transmission. Single nucleotide, Indel and VNTR polymorphisms in the *SLC6A4* gene have been implicated in the re-uptake efficiency; for example, the LL genotype of 5HTTLPR polymorphism is related with increased SERT concentration becoming the S carrier variants as a risk factor for psychiatric disorders [16–21,55].

**Table 3. Allelic frequencies reported in Latin American population.**

| Country | Trait | Cases | | | Controls | | | Reference |
|---|---|---|---|---|---|---|---|---|
| | | Sample size | S | L | Sample size | S | L | |
| Colombia | ASD | 105 | 0.59 | 0.41 | 171 | 0.52 | 0.48 | Present study |
| Colombia | Bipolar disorder | 103 | 0.49 | 0.51 | 112 | 0.53 | 0.47 | [Ospina-Duque et al., 2000] [37] |
| Colombia | MDD | 68 | 0.49 | 0.51 | 68 | 0.45 | 0.55 | [Pérez-Olmos, et al., 2016] [38] |
| Colombia | Bipolar disorder | 133 | 0.55 | 0.45 | 120 | 0.59 | 0.41 | [Ramos, et al., 2012] [39] |
| Colombia | MDD | 59 | 0.53 | 0.47 | 59 | 0.44 | 0.56 | [Escobar, et al., 2011] [40] |
| Mexico | Obsessive-compulsive disorder | 115 | 0.58 | 0.42 | 136 | 0.52 | 0.48 | [Camarena et al., 2001] [41] |
| Mexico | MDD | 104 | 0.52 | 0.48 | 335 | 0.60 | 0.40 | [Peralta-Leal et al., 2012] [42] |
| Mexico | ADHD | 78 | 0.43 | 0.57 | 56 | 0.55 | 0.45 | [Durán-González et al., 2018] [43] |
| Mexico | MDD and suicide attempt | 200 | 0.63 | 0.37 | 233 | 0.51 | 0.49 | [Sarmiento-Hernández et al., 2019] [44] |
| Argentina | MDD | 95 | 0.51 | 0.49 | 107 | 0.47 | 0.53 | [Cajal et al., 2012] [45] |
| Brazil | Bipolar Disorder | 167 | 0.37 | 0.63 | 184 | 0.36 | 0.64 | [Neves et al., 2008] [46] |
| Brazil | Schizophrenia* | 39 | 0.42 | 0.58 | 98 | 0.38 | 0.62 | [Mendes De Oliveira et al., 1998] [47] |
| Brazil | Bipolar Disorder* | 47 | 0.40 | 0.60 | 98 | 0.38 | 0.62 | [Mendes De Oliveira et al., 1998] [47] |
| Brazil | Dysthimia~ | 62 | 0.43 | 0.57 | 197 | 0.38 | 0.62 | [Oliveira et al., 2000] [48] |
| Brazil | Bipolar disorder~ | 64 | 0.38 | 0.62 | 197 | 0.38 | 0.62 | [Oliveira et al., 2000] [48] |
| Brazil | MDD~ | 66 | 0.42 | 0.58 | 197 | 0.38 | 0.62 | [Oliveira et al., 2000] [48] |
| Brazil | Anxiety | 129 | 0.43 | 0.57 | 96 | 0.43 | 0.57 | [Bortoluzzi et al., 2014] [49] |
| Brazil | Suicide in depressed patients | 84 | 0.48 | 0.52 | 152 | 0.44 | 0.56 | [Segal, et al, 2006] [50] |
| Brazil | Epilepsy | 175 | 0.47 | 0.53 | 155 | 0.45 | 0.55 | [Schenkel et al., 2011] [51] |
| Brazil | ASD | 151 | 0.44 | 0.56 | 179 | 0.45 | 0.55 | [Longo, et al, 2009] [52] |
| Brazil | Schizophrenia—Bipolar disorder | 99 | 0.40 | 0.60 | 60 | 0.47 | 0.53 | [Krelling et al., 2008] [53] |
| Brazil | Obsessive-compulsive disorder | 78 | 0.54 | 0.46 | 202 | 0.52 | 0.48 | [Meira-Lima et al., 2004] [54] |

ASD: Autism Spectrum Disorder, MDD: Major Depressive Disorder, ADHD: Attention Deficit Hyperactivity Disorder.

*Two traits evaluated independently but using the same control group,

~Three traits evaluated independently but using the same control group

The role of 5-HTTLPR polymorphism in the ASD pathophysiology evaluated through case-control studies and family-based association have reflected conflicting results. For instance, two studies suggested S allele as risk variant for ASD [15,16], while three meta-analysis did not find association [26–28]. We assessed this polymorphism through a case-control approach failing to find association between 5-HTTLPR and idiopathic ASD, and also through a family-based assessment that did not find a preferential transmission of either S or L allele. A previous evaluation of this polymorphism in Colombian individuals with ASD also failed to find a significant association [56].

Phenotypic heterogeneity in ASD has been also studied in relation with the 5-HTTLPR polymorphism, two reports suggested an association of the S allele with higher severity in

**Table 4. Results of meta-analysis.**

| Genetic model | Meta-analysis | | Heterogeneity | | Bias | | Recalculated OR with trim and fill method | |
|---|---|---|---|---|---|---|---|---|
| | Pooled OR (95% CI) | p Value | I² | p Value | Egger p value | Begg p value | Pooled OR (95% CI) | p Value |
| S vs L | 1.0698 [0.9880; 1.1584] | 0.0962 | 33.1% | 0.0673 | 0.5125 | 0.9775 | 1.0625 [0.9593; 1.1769] | 0.2306 |
| SS vs SL+LL | 1.1008 [0.9644; 1.2565] | 0.1548 | 19,5% | 0.2034 | 0.5359 | 0.7997 | 1.0913 [0.9355; 1.2730] | 0.2513 |
| LL vs SL+SS | 0.9146 [0.8066; 1.0371] | 0.1641 | 13,7% | 0.2766 | 0.636 | 0.8435 | 0.9218 [0.7975; 1.0654] | 0.2550 |

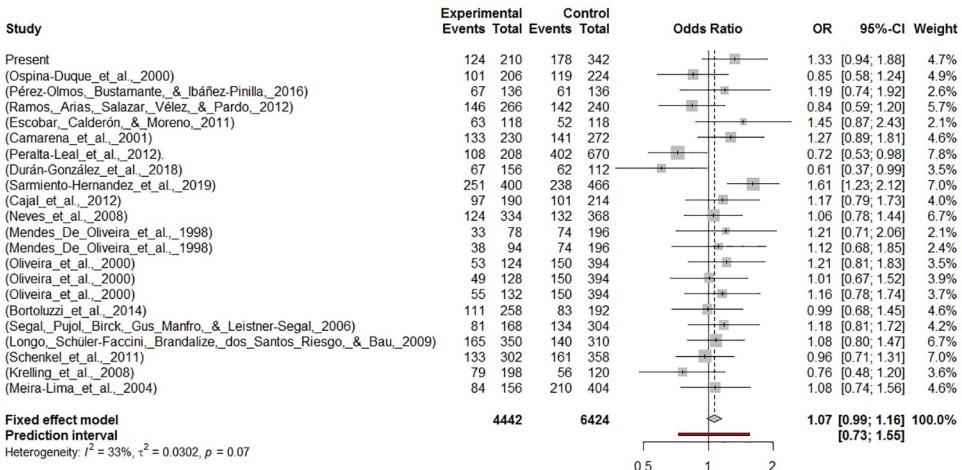

**Fig 1. Meta-analysis evaluating 5-HTTLPR polymorphism in Latin American population.** Forest plot for the allelic model. The trait and country for each study are found in Table 3. The forest plots for other two models are found in S2 Fig.

social communication impairment [57,58], while another group associated the severity of this trait with the LL genotype for 5-HTTLPR and the AA genotype for rs25531 [59]. Brune et al., 2006 also reported the LL genotype associated with severity of repetitive behaviors [58]. In our study, we did not find an association between 5-HTTLPR genotype and severity in the three core ASD symptoms (repetitive/restricted behaviors, impairment in social interaction and variable communication skills) just like other three reports [52,60,61]. An association between aggressive behaviors and the serotoninergic system has been also suggested [13,60,62,63]; in our study we observed that self-injury behaviors were mainly reported in individuals with the SS genotype (15/29), and although we did not find a significant association ($p>0.05$) our sample size does not have the statistical power to reveal this association. A summary of these conflicting results is presented in S7 Table.

Replication of results across ethnicities has been postulated as a good strategy to discover true risk variants or to understand if the lack of replication reflects the complexity in the genetic architecture of some complex disorders as ASD [64,65]. Although two meta-analyses recovering case-control studies mainly from European, American, and Asian population did not find association of this polymorphism with ASD, they also confirmed high heterogeneity in the included studies probably explained by population ethnicity [27,28]. These results suggest that the 5-HTTLPR polymorphism varies across ethnicities and the genetic background of each population may have a role in the risk conferred by this polymorphism [64,65].

Recognizing the under-representation of Latin American population in studies aimed to understand the role of 5-HTTLPR on worldwide [26,27,29], here we present a meta-analysis including articles which evaluate this polymorphism in Latin American individuals with psychiatric disorders as ASD, bipolar disorder, major depressive disorder, obsessive compulsive disorder, attention deficit hyperactive disorder, schizophrenia, dysthymia, anxiety disorder and suicide [37–54]. Although the S allele has been associated with increased risk for psychiatric disorders [16–21,55], our meta-analysis revealed no significant heterogeneity among studies, no publication bias and failed to find an association between 5-HTTLPR and a risk for psychiatric disorders. Comparing our frequencies with frequencies reported in other continents, the Latin American frequencies are more similar to those reported in Caucasian population [62,66–68], while S allele has been greater in Asian population [69,70] and L allele greater

in African population [70,71]. Nevertheless, none of Latin American studies had a strict genetic control for population substructure between cases and controls and according to previous reports the Brazilian population should have an over-representation of African component in some regions [72–74]. Even though Latin America presents a diverse population substructure, most of the regions presented in the meta-analysis are likely to have an elevated Caucasian component compared to native American and African backgrounds [70,75,76]; however, slight differences may not be discarded in some small or separate regions according to demographic history of these admixed countries [72–76].

In summary, our study supports the absence of association between ASD and 5-HTTLPR polymorphism in a homogeneous cohort of Colombian individuals with idiopathic ASD. Additionally, a meta-analysis evaluating this polymorphism in Latin American regions suggests that frequencies of short/long alleles in this under-represented population are relatively homogeneous to frequencies reported in Caucasian populations.

## Supporting information

**S1 Table. Genotyping data for each trio and ADOS score for each individual with idiopathic ASD.**
(DOCX)

**S2 Table. Genotyping data for each unrelated control.**
(DOCX)

**S3 Table. Results of case-control association test between ASD and 5-HTTLPR polymorphism.**
(DOCX)

**S4 Table. Transmission disequilibrium test in Colombian trios with ASD.** TDT was performed with families having heterozygotes parents (49 fathers and 42 mothers).
(DOCX)

**S5 Table. Phenotypic traits in individuals with ASD according to 5-HTTLPR genotypes.**
Two sub-categories were well-thought-out, patients who loss the acquired skills at 2 years old were classified as "ASD with regressive development" and patients with ASD and higher cognitive skills were classified as "High functioning ASD". Intellectual disability was evaluated during clinical consultation without standardized test. Aggressive behaviors were analyzed according to ADIR reports.
(DOCX)

**S6 Table. Sensitivity analysis.**
(DOCX)

**S7 Table. 5-HTTLPR polymorphism and ASD severity.** This table summarizes some results of studies evaluating severity of ASD symptoms and the serotoninergic system.
(DOCX)

**S1 Fig. Flowchart for selection of studies in the meta-analysis.**
(DOCX)

**S2 Fig. Meta-analysis evaluating 5-HTTLPR in Latin American population.** The trait and country for each study and are in Table 3. a. Forest plot for SS vs SL+LL model, b. Forest plot for LL vs SL+SS model.
(TIF)

**S3 Fig. Funnel plots to evaluate publication bias.** a. Funnel plot for S vs L model b. Funnel plot for SS vs SL+LL model, c. Funnel plot for LL vs SL+SS model.
(TIF)

**S4 Fig. Trim and fill funnel plots.** Funnel plots without missing studies a. for S vs L model, b. for SS vs SL+LL model and c. for LL vs SL+SS model.
(TIF)

**S1 Checklist. PRISMA checklist.**
(DOC)

## Acknowledgments

The authors are pleased to acknowledge the participant families and Instituto Colombiano del Sistema Nervioso Clínica Monserrat. Special thanks to Silvia Gonzalez Nieves, Daniela Castellanos and Camila Velasco from Universidad de Los Andes.

## Author Contributions

**Funding acquisition:** M. C. Lattig.

**Methodology:** D. L. Nuñez-Rios, R. Chaskel, A. Lopez, L. Galeano.

**Supervision:** M. C. Lattig.

**Writing – original draft:** D. L. Nuñez-Rios.

**Writing – review & editing:** M. C. Lattig.

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
