## [Decision Letter · Decision Letter 0]

12 May 2020

PONE-D-20-10849

The role of 5-HTTLPR polymorphism in Autism Spectrum Disorder and other psychiatric disorders in the Latin American population

PLOS ONE

Dear Dr Lattig,

Thank you for submitting your manuscript to PLOS ONE. After careful consideration, we feel that it has merit but does not fully meet PLOS ONE’s publication criteria as it currently stands. Therefore, we invite you to submit a revised version of the manuscript that addresses the points raised during the review process.

We would appreciate receiving your revised manuscript by Jun 26 2020 11:59PM. To enhance the reproducibility of your results, we recommend that if applicable you deposit your laboratory protocols in protocols.io, where a protocol can be assigned its own identifier (DOI) such that it can be cited independently in the future. For instructions see: http://journals.plos.org/plosone/s/submission-guidelines#loc-laboratory-protocols

We look forward to receiving your revised manuscript.

Kind regards,

Giuseppe Novelli

Academic Editor

PLOS ONE

Journal Requirements:

Reviewers' comments:

Reviewer's Responses to Questions

**Comments to the Author**

1. Is the manuscript technically sound, and do the data support the conclusions?

Reviewer #1: Yes

Reviewer #2: Yes

2. Has the statistical analysis been performed appropriately and rigorously? 

Reviewer #1: Yes

Reviewer #2: Yes

3. Have the authors made all data underlying the findings in their manuscript fully available?

Reviewer #1: Yes

Reviewer #2: Yes

4. Is the manuscript presented in an intelligible fashion and written in standard English?

Reviewer #1: Yes

Reviewer #2: Yes

5. Review Comments to the Author

Reviewer #1: In this paper, in light of the controversial evidence on the association between the 5-HTTLPR short variant (S) and 5-HTTLPR long variant (L) and ASD, the authors carried out a meta-analysis on the topic, focusing on the Colombian population.

The two variants fall into the promoter sequence of the SLC6A4 gene that codes for a transporter and regulator of serotonin levels, located in the presynaptic membrane.

The individuals were divided into two groups: the cases group consisting of 105 ASD trios, selected on the basis of the DSM-V, ADOS and ADI-R scale criteria; the control group of 171 apparently healthy unrelated individuals, from the same geographical area. After genotyping, the authors investigated the association between ASD and polymorphisms based on allelic, genotypic, dominant and recessive models. A linear regression analysis and association with quantitative traits was conducted to evaluate the association with severity for common traits for ASD. Subsequently, the results were compared with a meta-analysis on the data present in literature. In conclusion, the authors did not find a significant association between ASD and 5-HTTLPR polymorphisms in their case-control study. In addition, their meta-analysis highlighted a homogeneity of the frequency of the two variants in the Colombian population, which appears to be similar to that of the Caucasian population, but different from the African and Asian ones.

The article, despite its simplicity, is interesting and well-focused in the descriptive part. It would be appropriate to adapt the Discussion section accordingly, avoiding elements of repetitiveness, in order to make the reading more smooth and pleasant.

According to the Preferred Reporting Items for Systematic Review and Meta-analysis statement (PRISMA checklist 2009) we suggest revising the title to introduce a statement regarding meta-analysis work.

It might be useful to describe the information sources including databases’ dates of coverage and eventual contact with study authors. Aiming for a stronger significance of the results, we suggest to include any assessment of publication bias and to describe the methods used to investigate the presence of missing studies. We also recommend specifying whether duplicates have been found. If so, the authors might want to insert the duplicates exclusion step in the flow chart (S1 Figure) in the screening section.

In the text line 119, the information in the first parenthesis “(S vs )” seems to be incomplete.

In the text lines 130-133, we recommend the substitution of the term “synapsis” (that is the pairing of homologous chromosomes during the first prophase of meiosis) with the term “synapse”. We also recommend to rephrase the sentence from text line 132 to 133.

The main points are consistent with the analysis carried out. The tables are easy to interpret and the discussion does not highlight major issues.

However, the cohort under examination is reduced in number, therefore it cannot be considered statistically strong, as reported by the authors. The results could also be affected by stratification in the population under examination.

Reviewer #2: This study is a case-control analysis whose purpose is to evaluate a possible association between S/L 5HTTLPR polymorphism and idiopathic ASD in Colombian population. Moreover, they also investigated if there is a preferential transmission of the S or L allele and if there is association between the 5HTTLPR genotype and the severity of ASD symptoms, considering three main phenotypic traits (social communication, Restricted/repetitive behaviors and Social interaction) evaluated through ADOS scale. For a deeper analysis in comparison with the current scientific findings on this topic, three databases were consulted for a literature search: PubMed, ScienceDirect and Scielo. Among the results, 17 case-control studies fitted the inclusion criteria.

5-HTTLPR polymorphism was screened through PCR in 105 individuals with diagnosis of Idiopathic ASD and 171 unrelated and unaffected individuals belonging to the same region. The polymorphism was evaluated under the allelic (L VS S), genotypic (SS vs LS vs LL), dominant (LL/LS vs SS) and recessive (SS vs LS/LL) models. No significant association was found between 5-HTTLPR and ASD (after Bonferroni’s correction) [table 1]. The family-based association between the genotype and ASD in the 105 trios was evaluated with TDT test, resulting in no preferential transmission of either the S or L allele.

Regarding the association between ASD core symptoms severity and 5-HTTLPR according to genotype, none was found using statistical tests [table 2].

They also conducted a meta-analysis evaluating 5-HTTLPR polymorphism in Latin American population and its relation with psychiatric disorders[table 3; figure 1], resulting in no significant association.

As pointed out by the author, there are already three meta-analysis that demonstrate that the S/L 5HTTLPR polymorphism is not a risk factor for ASD, however this work has also taken into account the ethnical heterogeneity as a factor of influence. Through the meta-analysis they showed also that the frequency of L/S alleles is in the Latin-American population is quite similar to the reported frequency for Caucasian population (thus quite homogeneous). Due to conflicting results in other studies, it is yet to be confirmed if 5-HTTLPR polymorphism actually varies between ethnicities and if the genetic background of each population may have a role in the risk conferred by this polymorphism [27;28;29].

I think this work is interesting and able to offer a more comprehensive point of view about the relation between ASD and 5-HTTLPR polymorphism, especially of course in Latin American population.

All the tables presented are clear and easy to comprehend.

The principal Flaw I personally found in the article, was how the “Discussion” section was constructed.

Some topics were actually proposed more than once, rather than being articulated cohesively. For example, while explaining the preferential transmission of L or S alleles: lines 140-144 vs lines 176-178 (in association with ethnicity).

Some sentences are a little bit redundant and could have been better formulated, like: “Comparing our frequencies with… of the L allele” (lines: 194-198).

At line 119 there is a letter missing: (S vs L*).

6. PLOS authors have the option to publish the peer review history of their article (what does this mean?). If published, this will include your full peer review and any attached files.

Reviewer #1: No

Reviewer #2: No

---

## [Author Response · Author response to Decision Letter 0]

2 Jun 2020

We appreciate the careful reading and the provided feedback on the manuscript “The role of 5-HTTLPR polymorphism in Autism Spectrum Disorder and other psychiatric disorders in the Latin American population”. We changed the original title to “The role of 5-HTTLPR in autism spectrum disorder: New evidence and a Meta-Analysis of this polymorphism in Latin American population with psychiatric disorders”. We incorporated suggestions made by the reviewers and editor with track changes in the “Revised Manuscript with track changes” and answer them below:

A/ We adjusted the revised manuscript file according to PlosOne requirements.

We also included new tables (S1, S2 and S3 Tables) to fulfill publicly available information belonging to this manuscript. S1 Table includes genotypic information of trios and ADIR and ADOS scores for each individual with ASD. S2 Table has the genotypic information of controls. The S3 table includes the statistical results for each model.

2. PLOS requires an ORCID D for the corresponding author in Editorial Manager on papers submitted after December 6th, 2016. Please ensure that you have an ORCID iD and that it is validated in Editorial Manager. 

A/ We included the ORCID ID of the corresponding author as follows https://orcid.org/0000-0003-2113-9266

Additionally, while reviewing the prisma checklist we encountered an article that contained all the genetic data but was initially excluded because it did not contain Hardy-Weinberg information. With the information provided by the authors in the article it was possible for us to obtain the HW results and therefore include it in our study. The results with the new information did not change the overall results 

Reviewers' comments:

Reviewer's Responses to Questions

Comments to the Author

 Reviewer # 1: 

1. The article, despite its simplicity, is interesting and well-focused in the descriptive part. It would be appropriate to adapt the Discussion section accordingly, avoiding elements of repetitiveness, in order to make the reading more smooth and pleasant.

According to the Preferred Reporting Items for Systematic Review and Meta-analysis statement (PRISMA checklist 2009) we suggest revising the title to introduce a statement regarding meta-analysis work.

A/ We appreciate your comments, they are very helpful and aimed to improve the quality of our work. _The original tittle “The role of 5-HTTLPR polymorphism in Autism Spectrum Disorder and other psychiatric disorders in the Latin American population” was modified to “The role of 5-HTTLPR in autism spectrum disorder: New evidence and a Meta-Analysis of this polymorphism in Latin American population with psychiatric disorders” to include the meta-analysis performed 

2. It might be useful to describe the information sources including databases’ dates of coverage and eventual contact with study authors. Aiming for a stronger significance of the results, we suggest to include any assessment of publication bias and to describe the methods used to investigate the presence of missing studies. We also recommend specifying whether duplicates have been found. If so, the authors might want to insert the duplicates exclusion step in the flow chart (S1 Figure) in the screening section.

A/ We included the PRISMA checklist 2009 that contains information regarding dates, number of duplicated studies and authors contact in the paragraph of “Literature search”. Moreover, the paragraph “Statistical analysis” was modified to include the methodology used to evaluate for publication bias (Funnel plots, Egger’s / Begg’s tests) and the presence of missing studies (Trim and fill method). 

A sensitivity assessment was also included to evaluate a risk of bias within selected studies. 

The section Literature Search and Statistical Analysis was modified to include all these aspects. 

3.In the text line 119, the information in the first parenthesis “(S vs )” seems to be incomplete.

A/The L letter was added to correct this mistake. 

4. In the text lines 130-133, we recommend the substitution of the term “synapsis” (that is the pairing of homologous chromosomes during the first prophase of meiosis) with the term “synapse”. We also recommend to rephrase the sentence from text line 132 to 133.

A/We substituted the “Synapsis” by “Synapse” term. Moreover, we rephrased the next sentence for better understanding. 

Reviewer #2: 

1. I think this work is interesting and able to offer a more comprehensive point of view about the relation between ASD and 5-HTTLPR polymorphism, especially of course in Latin American population. All the tables presented are clear and easy to comprehend.

2. The principal Flaw I personally found in the article, was how the “Discussion” section was constructed. Some topics were actually proposed more than once, rather than being articulated cohesively. For example, while explaining the preferential transmission of L or S alleles: lines 140-144 vs lines 176-178 (in association with ethnicity). Some sentences are a little bit redundant and could have been better formulated, like: “Comparing our frequencies with… of the L allele” (lines: 194-198).

A/We appreciate your comments, they are very helpful and aimed to improve the quality of our work. We removed repetitive information previously consigned in these paragraphs and rephrased sentences to provide a better formulation of this information.

3. At line 119 there is a letter missing: (S vs L*).

A/The L letter was added to correct this mistake.

---

## [Editor Report · Decision Letter 1]

17 Jun 2020

The role of 5-HTTLPR in autism spectrum disorder: New evidence and a Meta-Analysis of this polymorphism in Latin American population with psychiatric disorders

PONE-D-20-10849R1

Dear Dr. Lattig,

We’re pleased to inform you that your manuscript has been judged scientifically suitable for publication and will be formally accepted for publication once it meets all outstanding technical requirements.

Kind regards,

Giuseppe Novelli

Academic Editor

PLOS ONE
---

## [Editor Report · Acceptance letter]

23 Jun 2020

PONE-D-20-10849R1 

The role of 5-HTTLPR in autism spectrum disorder: New evidence and a Meta-Analysis of this polymorphism in Latin American population with psychiatric disorders 

Dear Dr. Lattig:

I'm pleased to inform you that your manuscript has been deemed suitable for publication in PLOS ONE. Congratulations! Your manuscript is now with our production department. 

Kind regards, 

on behalf of

Prof. Giuseppe Novelli 

Academic Editor

PLOS ONE